# Use of Information and Communication Technology by South Korean Occupational Therapists Working in Hospitals: A Cross-Sectional Study

**DOI:** 10.3390/ijerph19106022

**Published:** 2022-05-16

**Authors:** Na-Kyoung Hwang, Sun-Hwa Shim, Hye-Won Cheon

**Affiliations:** 1Department of Occupational Therapy, Seoul North Municipal Hospital, Seoul 02062, Korea; occupation81@gmail.com; 2Department of Occupational Therapy, College of Medical Science, Jeonju University, Jeonju 55101, Korea; 3Department of Dental Hygiene, College of Health Science, Howon University, Gunsan 54058, Korea

**Keywords:** occupational therapy, information and communication technology, smart device, online survey

## Abstract

The convergence and development of information and communication technology (ICT) have brought changes to occupational therapy practices, posing novel challenges for occupational therapists (OTs). This study aimed to investigate current practices of ICT use and factors affecting the clinical use of ICT among Korean OTs. An online survey was conducted among 158 domestic OTs working in hospitals. Participants reported that the therapeutic use of ICT positively affected client outcomes, ICT choice, and continued use. Participants highlighted the necessity to assess the ability of clients to use smart devices and ensure familiarity in the OT process. Of respondents, 31% reported the application of ICT-based interventions or recommendations in clinical practice. The use of ICT was predominantly associated with cognitive function, leisure activities, and information access and communication. A significant difference in barriers to ICT use was observed between familiar users and non-users. Familiar users reported a lack of knowledge and training as major barriers, whereas non-users reported expensive products or technology. Ease of use and usefulness were facilitators of ICT use among familiar users. Information and training opportunities are required to promote ICT use by OTs, and the usefulness of ICT must be realized via client-centered, customized approaches.

## 1. Introduction

The convergence of healthcare and information communication technology (ICT) has raised interest in the digital healthcare industry at a time of a paradigm shift from treatment and provider-centered medical services to prevention and client-centered services [1]. The Republic of Korea planned to foster the healthcare industry as a new field of growth in the era of the 4th industrial revolution, with the aim of reducing medical costs and improving the quality of medical care via the convergence of healthcare and ICT [2]. Improved client experience of care, improved population health, and reduced costs of healthcare per capita are the main goals of the global healthcare system. The IHI (Institute for Healthcare Improvement) Triple Aim framework for healthcare in the United States aligns with this [3].

The domestic ICT-based healthcare industry is expected to expand steadily in the future due to the aging population and an increase in the number of chronic diseases, income levels, and interest in health [2]. Activities supporting healthcare cover a broad spectrum through advanced technologies, including information systems such as internet or web-based resources, management systems such as electronic health records, communication systems such as telecare and telemedicine, and decision support systems such as computerized clinical decision support [4]. In addition, the recent convergence of technologies such as bio-healthcare, artificial intelligence, big data, the Internet of things, and the cloud has enabled the easy access of healthcare at any time and from any location, beyond the existing healthcare field [5].

Occupational therapists (OTs) work with clients throughout their lifetimes from infancy to old age, with the common goal of promoting, developing, recovering, and maintaining the skills necessary to participate in their daily activities to prevent dysfunction and promote health [6]. The convergence and development of technologies have altered the practice of occupational therapy and the nature of the therapeutic relationship [7]. These changes have created new challenges for OTs in their profession, as they must be able to use existing and emerging ICT in line with these continual changes. The use of ICT for health is becoming a prominent field in which routines and innovative forms are utilized to address health-related needs [8]. ICT-based interventions, such as internet or web-based interventions, video conferencing, telehealth systems, and mobile applications, increase clients’ access to healthcare. ICT enables participation in activities of interest, improves self-confidence, and reduces social isolation [9]. OTs have used and recommended technology as a component of interventions and care for clients with various health conditions [10,11]. These technologies have been applied to various areas in occupational therapy, highlighting the effectiveness, perspective, and feasibility of ICT use [12,13,14,15]. Zonneveld et al. [12] demonstrated that ICT-based interventions improve client participation in everyday life in a cost-effective way. Collins [13] reported that OTs are primarily using assistive technology to address safety concerns in the dementia population. The OTs participating in the study made extensive use of high-tech devices, including Wii, iPads, iPhones, computers, and complex medication management systems, as well as low-tech devices such as memory walls. Other studies reported that ICT-based environmental assessments and interventions for clients who required home modifications have potential feasibility for improving client home safety awareness and facilitated collaboration with stakeholders in the home modification process [14,15].

ICT can provide solutions that improve quality of life and healthcare by promoting healthy lifestyles in line with the paradigm shift centered on disease prevention and management based on the existing healthcare paradigm of diagnosis and treatment [1].

Approximately 52% of OTs in South Korea work in rehabilitation settings in hospitals [16]. In particular, ICT such as smart devices and applications are rapidly being integrated in current rehabilitation settings, and novel functionalities and apps are improving device accessibility [17]. In this regard, OTs play a key role in the therapeutic application of technology and recommendations for clients, and their roles evolve in parallel with technological advancements [18]. However, no study to date has investigated the current status of ICT use in areas of occupational therapy, types of ICT applied for therapeutic use and recommended to clients, and factors influencing ICT use in Korea. Therefore, this study aimed to investigate current practices of Korean OTs working in hospitals with regard to ICT-based interventions and recommendations and to identify the factors that promote and hinder clinical application of ICT.

## 2. Methods

### 2.1. Survey Design

A cross-sectional online survey was conducted to identify the current practices of Korean OTs working in hospitals regarding ICT-based interventions and recommendations and to identify the factors influencing the clinical use of ICT by OTs. The survey was divided into four sections: (1) demographic profile (i.e., gender, age, clinical experience, areas of clinical practice, etc.), (2) familiarity with and use of ICT (i.e., possessed ICT-related knowledge and used it in practice) and OTs’ opinions on the therapeutic use of ICT in clinical settings (i.e., its impact on client’s occupational performance, technology choice and continuous use, and the necessity of integrating the client’s ICT familiarity into the OT process), (3) current ICT use reported by familiar users and frequency of use, and (4) factors influencing ICT use (i.e., facilitators and barriers to ICT use). In the third section, the areas of technologies used were broadly classified into personal factors, self-reliance and participation, and environment, which constitute key concepts of the Human Development Model-Disability Creation Process [19]. The domains and areas were recomposed as follows on the basis of the report of Aboujaoudé et al. [20]: personal factors, including cognitive function, communication, and health-related information; self-reliance and participation factors, including mobility, general planning and management of daily activities, leisure activities and information access, fall prevention, self-management or household activities, medication management, and prevention of burns and water damage; and environmental factors, including smart environment, telecare, and caregiver role support. The ICT lists for each area were composed based on supply items of the National Information and Communication Assistive Technology Devices of Korea [21] and National Classification System for Assistive Technology Device of Korea [22].

An Internet-based survey comprising 38 questions was created. For the questions about familiar users’ current ICT use and the factors influencing ICT use, the respondents were provided a text field in which they could provide their answers to the open-ended questions, as well as a list of choices for the categorical closed-ended questions. Thereafter, the responses to the open-ended questions were configured as answer choices and included in the analysis. The survey items were reviewed and revised by two OTs working in hospitals and one assistive technology specialist. Subsequently, three OTs working in hospitals performed a pilot assessment for item clarity, length, and face validity.

### 2.2. Participants

Occupational therapy subcommittee (society or alliance)-registered OTs were invited to participate in the study via email. Overall, 967 OTs were sent the invitation to participate, and 183 OTs opened the survey. Among the OTs who received the e-mail, those working in a hospital were asked to respond. The survey was deployed using the Google Survey web tool (https://www.google.com/intl/ko_kr/forms/about/) (accessed on 9 February 2022). The online survey was active for three weeks beginning on 11 February 2022. During the survey period, one survey invitation was sent by email to encourage participation. Prior to commencing the survey, online informed consent was obtained from the respondents, and the survey took approximately 10 min. The G*Power 3.1 program was used to determine the sample size, and as a result of analysis with effect size = 0.15, α = 0.05, and Power (1 − β) = 0.80, this study required a minimum sample size of at least 145 participants. Inclusion criteria were OTs working in hospital rehabilitation settings and OTs with more than one year of clinical experience (skilled in occupational therapy practice). Based on the levels of professional expertise defined by Benner [23] (novice, advanced beginner, competent, proficient, and expert), only therapists with a level of “advanced beginner” or higher and with more than one year of clinical experience were included. The exclusion criteria included OTs not practicing in hospital rehabilitation settings and novice OTs in their first year of practice after graduation.

### 2.3. Data Analysis

Only data from OTs who completed the survey in full were included. Statistical analyses were performed using SPSS version 15.0 (IBM Corp., Armonk, NY, USA). OTs who participated in this survey were classified into familiar users, familiar non-users, and unfamiliar non-users according to ICT-related knowledge and use in practice. “Familiar users” refer to respondents who are familiar with ICT supporting the occupational therapy of clients and use the technology in clinical practice. “Familiar non-users” refer to respondents who are familiar with the technology but do not use it. “Unfamiliar non-users” refer to respondents who are not familiar with (they do not know about ICT) and do not use ICT. Descriptive statistics were obtained to identify demographic profiles, frequency of ICT use by familiar users, and factors influencing ICT use by users and non-users among OTs familiar with ICT. From the open-ended questions, we collected additional data, which were analyzed as answer choices. The added answer choices were virtual reality software in the cognitive function area and bone conduction hearing aids in the communication area. Pearson’s chi-squared analysis was performed to assess differences in opinions on ICT use between users and non-users. As the expected frequency of <5 occupied 40% of the total cells, the significance probability was identified using Fisher’s exact test.

## 3. Results

### 3.1. Demographic Characteristics

Results of 16 surveys with incomplete responses were excluded, and nine surveys with skipped logic that affected the amount of information received were also excluded. A total of 158 respondents were included in the analysis. Demographic profiles are presented in Table 1. The OTs who participated in this survey had an average age of 31.8 years and 8.0 years of practical experience. Bachelor’s degree (55.1%) was the highest education level among participants. Of the three groups classified according to ICT familiarity, familiar users exhibited the oldest average age (34.7 years), longest practical experience (10.3 years), and highest percentage with an education level of bachelor’s degree (49.0%). Most of the respondents worked at a university/general hospital (36.1%) and semi-hospital setting (36.1%). Familiar users constituted the highest percentage working at a university/general hospital (38.8%). The provinces of employment with the highest response rates were Seoul (21.5%) and Gyeonggi (18.4%). The number of OTs who reported themselves as familiar users was also the highest in Seoul (26.5%) and Gyeonggi (28.6%). Practice areas with high responses were neurological, assessment, and cognitive/perceptual, which were reported in more than 50% of participants in each group. With regard to occupational therapy service clients, more than 50% of participants in all groups reported central nervous system (CNS) disorders and geriatric diseases.

### 3.2. Opinions on the Therapeutic Use of ICT

Differences of opinion on the therapeutic use of ICT according to the degree of familiarity and use of ICT were assessed. Regarding the positive effect of the therapeutic use of ICT on the changes in clients’ occupational performance, 93.8% of familiar users, 82.9% of familiar non-users, and 61.8% of unfamiliar non-users agreed with this effect. In addition, 93.9% of familiar users responded that the therapeutic use of ICT by OTs affects clients’ ICT choice and continued use; 75.6% of familiar non-users and 51.5% of unfamiliar non-users agreed. Both items showed statistically significant associations between users (all *p* < 0.001). Regarding the need to check clients’ ability to use and their familiarity with smart devices in the occupational therapy process, 98% of familiar users, 90.3% of familiar non-users, and 58.9% of unfamiliar non-users agreed with such a requirement; a significant association between ICT users was found (*p* < 0.001) (Figure 1).

### 3.3. Application of ICT by Familiar Users in Clinical Practice

Regarding personal factors, cognitive function support technology (95.9%) demonstrated the highest use, followed by communication (81.6%), and health-related information technology (77.6%). The most-used technology related to personal factors was voice recordings or text memo function on tablets or smartphones (95.9%). With regard to self-reliance and participation, leisure activities and information access-related technologies (83.7%) exhibited the highest use, followed by mobility (69.4%), and self-management and household activities (69.4%). In this domain, information access using computers, tablets, smartphones, mobile games, Nintendo Wii (77.6%), online grocery purchases (67.3%), and apps for directions and public transportation (53.1%), such as Naver Map and Kakao Map, were associated with a high frequency of use. Regarding environmental factors, more than 50% of users familiar with ICT used the technologies for smart environments and telecare or caregiver role support. Among these technologies, websites providing information for client care exhibited the highest frequency of use (46.9%), followed by smart home appliances and environmental control systems, such as automatic heating and lighting control systems (40.8%) (Table 2).

The distribution of ICT use frequency of familiar users is presented in detail in Figure 2. Among personal factors, the domain with a high frequency of use (five or more times a week) was cognitive function (20.4%), and more than 65% of users reported using it one to four times a week in all three domains of personal factors. Among lifestyle habits, leisure activities and information access (16.3%) exhibited the highest frequency (≥5/week), and more than 50% of users reported that they used technologies one to four times in the domains of leisure activities and information access and mobility. The domains with high non-use frequency were burn prevention, flood damage (75.5%), and fall prevention (65.4%). With regard to environmental factors, approximately 50% of users reported using the technology at least once a week in two domains of technology related to smart environments and telecare or caregiver role support.

### 3.4. Factors Influencing ICT Use by OTs

The facilitators and barriers to ICT use by OTs familiar with ICT are presented in Table 3. Among familiar users, the item with the highest response rate as an ICT use facilitator was ease of use (operation, manipulation) (73.5%), followed by usefulness (65.3%), reasonable purchase price and maintenance costs (65.3%), and easily obtainable product (51.0%). Lack of knowledge and training of the therapist (61.2%) was the item with the highest response rate as barriers to ICT use among familiar users, followed by expensive product or technology (55.1%), and lack of financial and administrative support in the workplace (50.0%). Among familiar non-users, the highest response rate was for expensive product or technology (65.9%), followed by lack of experience of clients in ICT (63.4%), and lack of financial and administrative support in the workplace (41.5%).

## 4. Discussion

The purpose of this survey was to investigate the current practices of ICT use among South Korean OTs working in hospitals. Only 31% of the respondents reported to be familiar with ICT. Lack of knowledge and training of the therapist was identified as the main factor that prevented the use of ICT by OTs. The results also suggested that easy-to-use operation for clients, usefulness when applied to the client, and reasonable purchase and maintenance costs promoted the therapeutic use of ICT.

### 4.1. Opinions on the Therapeutic Uses of ICT

Several studies have reported that ICT increases client participation and adherence to therapeutic activities, supports shared decision-making, and improves client’s outcomes [24,25,26]. Most of the OTs participating in this survey reported a positive influence of ICT. However, the survey revealed significant differences in opinions regarding the influence of ICT use by OTs on client outcomes and ICT choice and continued use according to the familiarity and use of ICT. Familiar users reported more positive opinions, reflecting how they felt about the usefulness of ICT, improved outcomes, and increased client engagement. Further, the need to assess the client’s ability to use smart devices and familiarity in the occupational therapy process were significantly different among groups, with all groups exhibiting moderate-to-high ratings. OTs should thus determine the candidacy and appropriateness of ICT on a case-by-case basis using clinical judgment after considering several factors, such as client-associated factors and activity requirements, performance skills and patterns, context and environment, and variability in these factors [27]. This may have underpinned the responses of OTs indicating the need to understand the client’s familiarity with ICT as a client-related factor in order to determine the appropriate ICT and application level.

### 4.2. Familiarity with and Use of ICT

Of the total respondents, 57% (*n* = 90) reported that they were familiar with ICT, but only 31% (*n* = 49) used it therapeutically. Integrating ICT into the rehabilitation process can support clients’ social participation [28,29] and contribute to improving their quality of life [30,31]. Healthcare professionals play a key role in providing appropriate ICT solutions to clients and caregivers [24,32]. Healthcare professionals are responsible for recommending appropriate ICT to clients and providing support in the method of using ICT and how to address issues that may arise when using ICT, such as updating [33,34]. However, the results of this survey revealed a gap between the performance role required for ICT use and the knowledge and usage practices of ICT by domestic OTs. Despite the changes in healthcare paradigms due to the development of ICT and the therapeutic use and positive effects of ICT reported in various studies [24,29,31], various aspects of ICT utilization in the current domestic practice of OTs are not clearly integrated. Although the potential of ICT as a therapeutic tool and the underlying evidence for its clinical application need to be actively addressed by clinicians, the current opportunities for continuing ICT education or ICT knowledge acquisition are insufficient. Therefore, opportunities for active education or knowledge acquisition and sharing are necessary to promote the dissemination of the latest knowledge and clinical cases of ICT use in the future.

OTs working in hospitals in South Korea reported the use of various technologies despite the low ICT usage rate. The area with the highest usage rate was cognitive function in the personal factor domain, followed by leisure activities and information access in the domain of self-reliance and participation, and communication in the personal factor domain. In terms of neurological rehabilitation, the field of cognitive rehabilitation for cognitive dysfunction caused by neurological disorders is continuously expanding [35]. Cognitive rehabilitation is one of the core tasks of domestic OTs and is a therapeutic area with high importance and frequency of performance [36]. In line with this, the observation that familiar users of this survey reported various ICT applications and high frequency of use in the cognitive function area is considered to reflect the role and high degree of professionalism of domestic OTs in cognitive rehabilitation.

In relation to leisure activities and information access, certain populations, especially the elderly, may not be able to enjoy leisure activities because of the lack of financial leeway or information. In addition, older individuals may have difficulties in acquiring information because they may be unable to find the desired information amid the vast amount of available information. As for others, they may have the ability to find information, but they may face difficulties in finding customized information that suits them [37]. As such, addressing the challenges faced by the elderly in information access and acquisition and in participation in leisure activities can help this group acquire new knowledge and leisure skills through appropriate guidance and the clinical application of ICT. This aspect was reflected in the results of this survey. The familiar users reported a high frequency of leisure activities and information access using computers, tablets, and smartphones; mobile games; and console-based virtual reality programs. The high frequency of use in this area highlights the importance of using ICT and the usefulness of interventions in this domain.

Communication-related technologies support various needs and functions of oral communication, written information access, and emotional communication. To implement this, various technologies such as ICT devices, software, digital content, smart media platform services, the Internet of things, and public convergence services are currently used [38]. However, users demonstrated a biased use of oral communication technologies and certain technologies of written information access. Most respondents used devices and app-based technologies. In the future, it will be necessary to investigate the clinical application and usefulness of various technologies that support and improve the communication of clients receiving OT services.

### 4.3. Factors Influencing ICT Use

The Internet, telehealth, digital devices that support self-management, and many other apps are inevitably becoming commonplace in the healthcare field [39]. However, healthcare professionals tend to underuse potential ICT resources [40]. Kapadia et al. [41] reported the presence of several problems such as trust in technology or lack of technical skills, costs related to ICT use, and difficulties in using ICT in the actual adoption of ICT by healthcare professionals. In this survey, familiar users and non-users generally exhibited similar responses regarding barriers to ICT use in clinical practice but demonstrated different responses regarding major barriers. Familiar users perceived a lack of knowledge and training of the therapist and expensive products or technology as major barriers. In contrast, non-users exhibited the highest response rate to expensive products or technology and lack of experience of the client in ICT as major barriers. These responses were derived from user experience lacking familiarity with ICT, and actual users more strongly perceived a lack of personal knowledge and need for training to use. Although ICT use and implementation is being adopted as an accredited occupational therapy program in some schools, it is not fully integrated into occupational therapy education. However, digital occupational therapy has recently become a topic of interest, and online seminars and conferences have been held by the Korea Association of Occupational Therapists [42]. OTs have an ethical obligation to remain updated with the latest technologies in clinical practice, and education and training are necessary to fulfill this requirement. Occupational therapy education programs need to keep pace with technological advances and provide comprehensive education on the therapeutic use of ICT to fulfill ethical obligations. In addition, it is necessary for OTs to become leading professionals who participate in, encourage, and promote the therapeutic use of technology in their workplace.

Users reported that the main facilitators were ease of use (operation, manipulation) and usefulness. The characteristics of easy operation are key factors for the adoption of ICT in rehabilitation by therapists [43]. Schaper et al. [44] reported that learning how to use technology takes a considerable amount of time, and a technology that is perceived as uncomplicated or easy to use has a positive effect on the intention to use it. With regard to the usefulness of ICT, previous studies reported that ICT such as apps should not be implemented in all clients despite their value as a therapeutic intervention tool [45,46]. Therefore, the observation that ICT usefulness was cited as the main facilitator of ICT use in this survey highlights the importance of using ICT with a person-centered approach in consideration of the client’s acceptability of ICT.

### 4.4. Limitations

The results of this study should be generalized with caution. The small sample size of this survey may not fully represent all domestic OTs working in hospitals. In addition, self-report questionnaires and voluntary participation of respondents may be subject to self-selection bias. In order to compose the ICT list, we referred to the ICT-related list proposed by institutions of assistive technology devices in Korea, which have been reviewed, revised, and pilot-tested by several experts. However, there may be limitations in the list composition. In addition, ICT lists and weekly frequency of use reported by respondents’ recall may be subject to information bias. Finally, the distribution of respondents according to employment area did not exhibit any notable differences, but differences in the work environment related to ICT by region may have affected the results of this survey.

## 5. Conclusions

The purpose of this study was to investigate the ICT use of domestic OTs working in hospitals. Changes in healthcare paradigms due to the development of ICT and the positive influence of the therapeutic use of ICT on client outcomes provide novel opportunities for therapeutic approaches. OTs who participated in this survey reported that the application of ICT affected the client’s occupational performance improvement and continuous ICT use. Further, they reported that it was necessary to confirm the familiarity of the clients with ICT in the OT process to ensure proper ICT adoption and application. This indicates that domestic OTs generally accept the use of ICT as an effective and useful therapeutic tool. ICT users of this survey reported ICT use in various areas; among them, cognitive function, leisure activities, information access, and communication technologies were the most used. However, except for these three areas, the frequency of weekly use was low. To promote the use of ICT in clinical practice, it is necessary to develop information-sharing media and educational programs based on currently applicable technologies and methods of use. Although ICT use may not positively impact all clients, an individualized and customized approach that considers the client’s ICT acceptability will permit the realization of the usefulness and potential value of ICT. Further research is needed to address the factors hindering the clinical use of ICT and to promote its use. In addition, it will be necessary to develop and commercialize technologies that facilitate clients’ access to and utilization of ICT in clinical practice.

## Figures and Tables

**Figure 1 ijerph-19-06022-f001:**
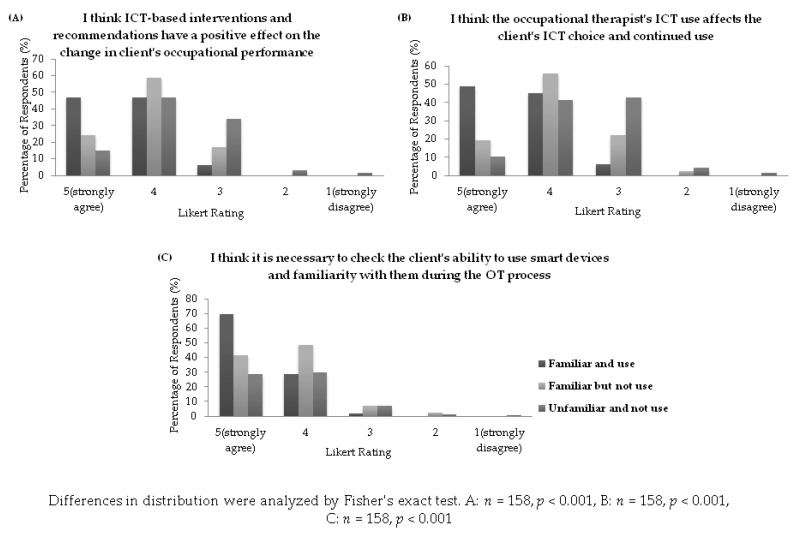
Distribution of opinion according to ICT use.

**Figure 2 ijerph-19-06022-f002:**
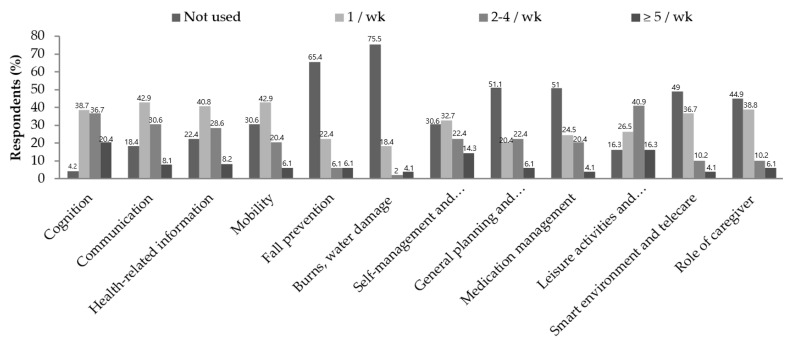
Frequency of ICT use by users.

**Table 1 ijerph-19-06022-t001:** Demographic profile.

Variables	Total = 158 *n* (%)	Familiar Users = 49	Familiar Non-Users = 41	Unfamiliar Non-Users = 68
Gender				
Female	90 (57.0)	19 (38.8)	23 (56.1)	48 (70.6)
Male	68 (43.0)	30 (61.2)	18 (43.9)	20 (29.4)
Age				
20–29	64 (40.5)	9 (18.4)	18 (43.9)	37 (54.4)
30–39	74 (46.8)	27 (55.1)	21 (51.2)	26 (38.2)
>40	20 (12.7)	13 (26.5)	2 (4.9)	5 (7.4)
Mean ± SD	31.8 ± 5.9	34.7 ± 6.2	31.1 ± 4.9	30.2 ± 5.5
Practice experience				
1–5 years	55 (34.8)	8 (16.3)	15 (36.6)	32 (47.0)
6–10 years	63 (39.9)	21 (42.9)	20 (48.8)	22 (32.4)
10–15 years	25 (15.8)	10 (20.4)	5 (12.2)	10 (14.7)
>10 years	15 (9.5)	10 (20.4)	1 (2.4)	4 (5.9)
Mean ± SD	8.0 ± 4.8	10.3 ± 4.5	6.8 ± 3.8	7.0 ± 4.7
Education				
Associate’s	28 (17.7)	6 (12.2)	8 (19.5)	14 (20.6)
Bachelor’s	87 (55.1)	24 (49.0)	22 (53.6)	41 (60.3)
Master’s & higher	43 (27.2)	19 (38.8)	11 (26.9)	13 (19.1)
Place of work				
University/General Hospital	57(36.1)	19 (38.8)	12 (29.3)	26 (38.2)
(Semi) Hospital	57(36.1)	17 (34.7)	15 (36.6)	25 (36.8)
Clinic	1 (0.6)	1 (2.0)	0 (0.0)	0 (0.0)
Nursing hospital	7 (4.4)	1 (2.0)	2 (4.9)	4 (5.9)
Public healthcare center (including dementia center)	36 (22.8)	11 (22.4)	12 (29.2)	13 (19.1)
Province of employment				
Seoul	34 (21.5)	13 (26.5)	7 (17.1)	14 (20.6)
Gyeonggi	29 (18.4)	14 (28.6)	5 (12.2)	10 (14.7)
Jeolla	20 (12.7)	5 (10.2)	6 (14.6)	9 (13.2)
Gyeongsang	27 (17.1)	4 (8.2)	10 (24.4)	13 (19.2)
Gangwon	27 (17.1)	3(6.1)	8 (19.5)	16 (23.5)
Chungcheong	21 (13.3)	10 (20.4)	5 (12.2)	6 (8.8)
Practice areas				
Assessment	110 (69.6)	33 (67.3)	29 (70.7)	48 (70.6)
Neurological	113 (71.5)	36 (73.5)	27 (65.9)	50 (73.5)
Musculoskeletal	51 (32.3)	21 (42.9)	8 (19.5)	22 (32.4)
Psychosocial	20 (12.7)	6 (12.2)	5 (12.2)	9 (13.2)
Prevocational/vocational	10 (6.3)	5 (10.2)	3 (7.3)	2 (2.9)
Cognitive/Perceptual	88 (55.7)	24 (50.0)	20 (48.8)	44 (64.7)
Hand therapy	31 (19.6)	12 (24.5)	7 (17.1)	12 (17.6)
Health promotion and wellness (community)	31 (19.6)	14 (28.6)	7 (17.1)	10 (14.7)
Developmental	33 (20.9)	12 (24.5)	4 (9.8)	17 (25.0)
Geriatric	72 (45.6)	20 (40.8)	22 (53.7)	30 (44.1)
Client				
CNS disorders	133 (84.2)	40 (81.6)	36 (87.8)	57 (83.8)
Musculoskeletal disorders	63 (39.9)	21 (42.9)	12 (29.3)	30 (44.1)
Cardiopulmonary disease	18 (11.4)	9 (18.4)	3 (7.3)	6 (8.8)
Hand injury	37 (23.4)	12 (24.5)	7 (17.1)	18 (26.5)
Arthritis and rheumatoid	35 (22.2)	10 (20.4)	7 (17.1)	18 (26.5)
Cancer	19 (12.0)	8 (16.3)	2 (4.9)	9 (13.2)
Geriatric diseases	105 (66.5)	24 (50.0)	29 (70.7)	52 (76.5)
Visual impairment	23 (14.6)	4 (8.2)	3 (7.3)	16 (23.5)
Hearing impairment	14 (8.9)	3 (6.1)	3 (7.3)	8 (11.8)
Mental illness	20 (12.7)	5 (10.2)	6 (14.6)	9 (13.2)
Intellectual disability	42 (26.6)	10 (20.4)	7 (17.1)	25 (36.8)
Cerebral palsy	51 (32.3)	13 (26.5)	9 (22.0)	29 (42.6)
Developmental disabilities	39 (24.7)	12 (24.5)	4 (9.8)	23 (33.8)
Genetic disorder	27 (17.1)	9 (18.4)	3 (7.3)	15 (22.1)
Learning disabilities	20 (12.7)	8 (16.3)	3 (7.3)	9 (13.2)
Language disorder	22 (14.0)	5 (10.2)	6 (14.6)	11 (16.2)
Autism	29 (18.4)	6 (12.2)	6 (14.6)	17 (25.0)

**Table 2 ijerph-19-06022-t002:** Percentages of ICT applied by familiar users (*n* = 49).

ICT-Based Interventions Or Recommendations	Users, *n* (%)
Personal factors	
Cognitive function	47 (95.9)
Voice recordings or text memo function on tablets or smartphones	47 (95.9)
Apps for cognitive function improvement training	38 (77.6)
App-based games (e.g., Baduk, Korean Chess, Sudoku, RumiCube)	38 (77.6)
Reminders app	11 (22.4)
Calendar app	11 (22.4)
Timer app	10 (20.4)
Photos app	9 (18.4)
Digital photo frame	3 (6.1)
Virtual reality software	1 (2.0)
Communication	40 (81.6)
Internet video call app or video phone	23 (46.9)
Voice recognition (voice command) function on smartphones	20 (40.8)
Smart AAC for communication	16 (32.7)
Image dictionary app on tablets or smartphones	16 (32.7)
Use of social media (e.g., blog, SNS, Kakao Talk open chat)	14 (28.6)
Text-to-Speech app or device	13 (26.5)
Smart AAC for language training	11 (22.4)
Adaptive smartphone (e.g., smartphone customized for the elderly)	9 (18.4)
Special mouse or special keyboard, key guard	7 (14.3)
Optical character reader	3 (6.1)
Braille translation app	2 (4.1)
Bone conduction hearing aids	1 (2.0)
Health-related information	38 (77.6)
Exercise program websites or video channels	35 (71.4)
Websites or video channels to obtain information about diseases and conditions	25 (51.0)
Apps to track physical activity (e.g., steps, repetitive movements)	22 (44.9)
Apps to record or track physiological changes (e.g., heart rate, blood pressure)	10 (20.4)
Apps to manage lifestyle patterns (e.g., drinking, smoking, exercise)	10 (20.4)
Apps to record or track psychological states (e.g., mood, anxiety, panic)	8 (16.3)
Web forums dealing with health-related topics	6 (12.2)
Self-reliance and participation	
Mobility	34 (69.4)
Apps for directions and public transportation (e.g., Naver Map, Kakao Map)	26 (53.1)
GPS location-tracking apps or watches	13 (26.5)
GPS white cane	4 (8.2)
Products for motion restriction (e.g., beds with motion sensor, motion detection alarms)	3 (6.1)
Fall prevention	17 (34.7)
Fall detection watches or bands	15 (30.6)
Personal emergency alarm systems	11 (22.4)
Night sensor light that detects motion	10 (20.4)
Prevention of burns or water damage	12 (24.5)
Water temperature indicator	9 (18.4)
Automatic hot water control system (e.g., automatic hot water control valve)	7 (14.3)
Leakage, flood detection alarm	3 (6.1)
Self-management and household activities	34 (69.4)
Online grocery purchases	33 (67.3)
Recipe websites or apps	18 (36.7)
Robot vacuum cleaner	16 (32.7)
Apps for household ledgers or budget planning	15 (30.6)
Digital Cooking Timer	12 (24.5)
Iron with automatic power off function	8 (16.3)
Apps for meal planning or organization	7 (14.3)
Sleep cycle monitoring apps	5 (10.2)
General planning and management of daily activities	24 (49.0)
Goal setting and management apps	23 (46.9)
Apps that record accomplished activities (e.g., Logbook)	15 (30.6)
Medication management	24 (49.0)
Medication reminder apps	19 (38.8)
Automatic pill dispenser	12 (24.5)
Drug search, prescription information management apps	7 (14.3)
Leisure activities and information access	41 (83.7)
Information access using computers, tablets, smartphones, mobile games, Nintendo Wii	38 (77.6)
Remote controls for the elderly (e.g., large button, universal)	11 (22.4)
Environment	
Smart environments and telecare	25 (51.0)
Smart home appliances and environmental control systems (automatic heating and lighting control systems)	20 (40.8)
Home CCTV	16 (32.7)
Remote control of home appliances and environment settings (e.g., room temperature, lighting, front door) via tablets and smartphones	14 (28.6)
Emergency pager (carried by the patient and connected to the phone in case of an emergency)	9 (18.4)
Telecare system	3 (6.1)
Caregiver role support	27 (55.1)
Websites providing information for client care	23 (46.9)
Devices that measure and monitor parameters such as blood pressure, oxygen saturation, and heart rate	13 (26.5)
Video conferences with caregiver	10 (20.4)

**Table 3 ijerph-19-06022-t003:** Facilitators and barriers to ICT use in clinical practice among familiar OTs (*n* = 177).

	Users (*n* = 49)	Non-Users (*n* = 41)
Facilitators, *n* (%)		
Ease of use (operation, manipulation)	36 (73.5)	-
Usefulness when applied to client	32 (65.3)	-
Reasonable purchase price and maintenance costs	31 (63.3)	-
Easily obtainable product	25 (51.0)	-
Financial and administrative support in the workplace	15 (30.6)	-
Therapist’s proficiency in using ICT	14 (28.6)	-
Reliability of the product or technology	9 (18.4)	-
Barriers, *n* (%)		
Lack of knowledge and training of therapist	30 (61.2)	14 (34.1)
Expensive product or technology	27 (55.1)	27 (65.9)
Lack of financial and administrative support in the workplace	24 (50.0)	17 (41.5)
Lack of experience of client in ICT	22 (44.9)	26 (63.4)
Lack of information about ICT within the department	17 (34.7)	16 (39.0)
Client’s negative attitude toward ICT application	14 (28.6)	11 (26.8)
Increase in therapist’s workload for ICT use	9 (18.4)	10 (24.4)
Negative experience of therapist in ICT application	6 (12.2)	2 (4.9)

## Data Availability

The data that support the findings of this study are available from the corresponding author upon reasonable request.

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
