# Peer review of "Use of Information and Communication Technology by South Korean Occupational Therapists Working in Hospitals: A Cross-Sectional Study"

_ijerph, 2022, doi:10.3390/ijerph19106022_

Round 1

Reviewer 1 Report

Thank you for having the opportunity to read your manuscript. I find it interesting to present how ICT technology is used among OTs in South Korea. However, the result needs to be better discussed from a general point of view in line with the scope of the journal. I also think the manuscript can benefit from reflecting on the in South Korean OTs representativeness to international OT and health care.

Regarding the included participants, it should be presented how many OTs initially were addressed with the survey. Thereby the number of non-respondents should be presented in the result section (or the percentage of respondents) and it should be reflected on in the discussion section. This is an important information in relation to generalisability that is mentioned in the discussion about limitations, but it can be better regarded in the method and result sections. - Was it mainly OTs familiar to ICT who answered the survey?

I find the description of the survey unclear. How many questions/items were in the survey? It seems like the survey include several open-ended questions with free text. How was this data analysed?

The result is descriptive and statistically differences between groups are explored. I think parts of this section could be rewritten to make the result more accessible. In general, each part of the result in presented by a number of examples. Can this be condensed/concluded better? Analysed?

The approach in the discussion section can be more generalisable. The reasoning about the South Korean context for OTs and OT education, can be expanded to a more deliberately international perspective and in relation to digital health? 

The title. The term” technology” is much broader than the specific information and communication technology that is the focus for the paper. I suggest it is defined all ready in the title.

The examples given on line 44-45 of what ICT may be is again very broad. Does this study really cover this variation? The examples refer to both systems used for management as well as for assessments and intervention and by clients themselves. Has any delimitation regarding technology been made? If not, would that be appropriate?

Line 61. It is written ”… provide clients with a connection to the outside world”. What do outside world mean, here? Consider rewriting.

Method section
How many items in total, did the survey comprise?

Line 104. It would be clearer and easier to follow the description of the survey if the domains and areas in the third section were not numbered by Arabic numbers (as section 1-4).

Line 114. “Most of the items comprised categorical closed items, and open text space was provided for the majority of the items to enable additional information … “ - How much text data was collected?? - How was the text in the open text space, analysed?? This should be described under the analysis section starting on line 140.

Line 124. The number of possible participants needs to be addressed to be able to describe the number of participants in relation to non-respondents.

Line 131. Again, it was calculated that the study required 145 participants. What did this mean in relation to the possible number of respondents? 

Line 137. The exclusion criteria; other professions? Was the survey not only referred to OTs??

What was the rationale behind detecting in which province the respondents worked? This information is not used further in the analysis and just mentioned in the discussion.

Results

Line 152. Nine surveys with skipped logic that affected the amount of information received, were excluded. What does this mean? What did these responses regard? Please explain what “affected the amount of information received” mean?

Table 1. The ratio between men and women participating was 57/43 %. How does this reflect the general population of OTs in South Korea? Any comments in the discussion?

I suggest the tables are presented according to general format with a left margin.

I find Figure 2 quite complex and thereby not that illustrative. I recommend it to be excluded and to add some of the information in written text.  

Discussion

In this part a new and extended aim is introduced. Again, what was the rationale between detecting the regions? It was not in the aim presented earlier in the paper. I suggest it is removed/rewritten.

Line 231. The information “31% of the respondents currently using ICT ...” is not clear where it is found in the result section.

Author Response

Regarding the included participants, it should be presented how many OTs initially were addressed with the survey. Thereby the number of non-respondents should be presented in the result section (or the percentage of respondents) and it should be reflected on in the discussion section. This is an important information in relation to generalisability that is mentioned in the discussion about limitations, but it can be better regarded in the method and result sections. - Was it mainly OTs familiar to ICT who answered the survey?

Response: The suggestions were further described in the Methods section.

I find the description of the survey unclear. How many questions/items were in the survey? It seems like the survey include several open-ended questions with free text. How was this data analysed?

Response: An Internet-based survey including 38 questions was created. In the questions about current ICT use reported by familiar users and factors influencing ICT use, opened text space to fill in when the respondent does not have the desired answer choice, along with categorical closed answer choices. After that, opened text answers were additionally configured as answer choices and included in the analysis. We described it further in the Methods section.

The result is descriptive and statistically differences between groups are explored. I think parts of this section could be rewritten to make the result more accessible. In general, each part of the result in presented by a number of examples. Can this be condensed/concluded better? Analysed?

Response: As suggested, the results were additionally presented numerically in ‘Opinions on the therapeutic use of ICT’, which described the comparison between groups according to respondents’ familiarity and use.

The approach in the discussion section can be more generalisable. The reasoning about the South Korean context for OTs and OT education, can be expanded to a more deliberately international perspective and in relation to digital health? 

Response: We further described what you suggested in 4.3. Factors influencing ICT use of the Discussion.

The title. The term” technology” is much broader than the specific information and communication technology that is the focus for the paper. I suggest it is defined all ready in the title.

Response: As suggested “technology” was changed to “information and communication technology” in the title. 

The examples given on line 44-45 of what ICT may be is again very broad. Does this study really cover this variation? The examples refer to both systems used for management as well as for assessments and intervention and by clients themselves. Has any delimitation regarding technology been made? If not, would that be appropriate?

Response: It seems that this description can be accepted as ICT domains according to technology classification. We have tried to describe the broad spectrum of activities supported by ICT in healthcare. So, line 44-45 was modified. In addition, these activities were included in the ICT interventions or recommendations presented in this survey.

Line 61. It is written ”… provide clients with a connection to the outside world”. What do outside world mean, here? Consider rewriting.

Response: As suggested, we modified the sentence.

Method section

How many items in total, did the survey comprise?

Response: The number of survey items was additionally described.

Line 104. It would be clearer and easier to follow the description of the survey if the domains and areas in the third section were not numbered by Arabic numbers (as section 1-4).

Response: The description composed of Arabic numerals in the domains and areas has been modified.

Line 114. “Most of the items comprised categorical closed items, and open text space was provided for the majority of the items to enable additional information … “ - How much text data was collected?? - How was the text in the open text space, analysed?? This should be described under the analysis section starting on line 140.

Response: From the questions provided with an opened text space we collected additional data, which were analyzed as one of the answer choices. The added answer choices were virtual reality software in the cognitive function area and bone conduction hearing aids in the communication area.

As suggested, it is further described in the Data analysis section.

Line 124. The number of possible participants needs to be addressed to be able to describe the number of participants in relation to non-respondents.

Response: As suggested, we further described it.

Line 131. Again, it was calculated that the study required 145 participants. What did this mean in relation to the possible number of respondents? 

Response: It means the minimum number of respondents expected to get any kind of meaningful result. If you think additional descriptions are needed in this regard, please comment. It will be reflected in the 2nd revision.

Line 137. The exclusion criteria; other professions? Was the survey not only referred to OTs??

Response: We have deleted “other professions”.

What was the rationale behind detecting in which province the respondents worked? This information is not used further in the analysis and just mentioned in the discussion.

Response: The results of the analysis on province of employment were briefly described in the Demographic characteristics of the Results.

Results

Line 152. Nine surveys with skipped logic that affected the amount of information received, were excluded. What does this mean? What did these responses regard? Please explain what “affected the amount of information received” mean?

Response: It means that the respondent did not report himself as a ‘familiar user’ but responded to ICT applied items or, conversely, reported himself as a familiar user but did not respond to most ICT applied items. (Despite the guidance that only 'Familiar User' respondents should respond to ICT applied items)

Table 1. The ratio between men and women participating was 57/43 %. How does this reflect the general population of OTs in South Korea? Any comments in the discussion?

Response: Regarding the representation of OTs in South Korea, the gender ratio is not considered to be an important point. Even in occupational therapy statistical information announced by the Korean Occupational Therapy Association, the male to female ratio cannot be confirmed.

I suggest the tables are presented according to general format with a left margin.

Response: As suggested, we modified it.

I find Figure 2 quite complex and thereby not that illustrative. I recommend it to be excluded and to add some of the information in written text.  

Response: Thank you for your comments. Since it may not be easy to understand information through the figure, a percentage value was entered above each bar. Please check and provide additional comments. We will reflect it in the 2nd revision.

Discussion

In this part a new and extended aim is introduced. Again, what was the rationale between detecting the regions? It was not in the aim presented earlier in the paper. I suggest it is removed/rewritten.

Response: Reference to 'regions' has been removed as suggested.

Line 231. The information “31% of the respondents currently using ICT ...” is not clear where it is found in the result section.

Response: The sentence was revised to “Only 31% of respondents were reported as 'familiar users' to ICT.”

* The revised parts were marked in red.

Author Response

Use of Technology by South Korean Occupational Therapists Working in Hospitals: A Cross-Sectional Study Throughout the paper use the terms familiar users and non-users throughout the paper; some sentences have “users”. The paper does not adequately address a distinction between Familiar non-users and Unfamiliar non-users.

Response: The terms of familiar users, familiar non-users, and unfamiliar non-users were mentioned in the data analysis of the method. Accordingly, 'users' means only familiar users. The term 'non-users' can be distinguished from 'familiar non-users' and 'unfamiliar non-users' because there is a mention of "... among familiar OTs" or "...familiar with ICT" in the sentence. Therefore, it is thought that there will be no confusion in understanding non-users terminology.

Discussion Section: 4.1. Opinions on the therapeutic uses of ICT 4.2. Familiarity with and use of ICT 4.3. Factors influencing ICT use Perhaps you need a distinct section in the discussion on Application in clinical practice that is mentioned in the results section 3.3.. Application is discussed in opinions, familiarity, and factor; however, having a distinct section is consistent with the results.

Response: Thanks for the comment. I may not have understood your comment correctly, but key findings from '3.3. Application ...' in Results was discussed in '4.2. Familiarity with and use of ICT'  in Discussion. There is a limit to be addressed all the results of the survey in discussion, and I don't think it is necessary to cover all the results. Please consider your comment. However, if you comment again that you need additional sections and related skills as you mentioned, we will reflect them in the 2nd revision.

Introduction

53 – recommend replacing cope with “engage” or “participate” in daily activities

Response: As suggested, we revised it.

59 - 61 – “ICT-based interventions, such as internet or web-based interventions, video conferencing, telehealth systems, and mobile applications, provide clients with a connection to the outside world.”  This paragraph seems to be discussing OT, ICT, and healthcare. This sentence begins with “ICT-based interventions…” which gives the reader the impression these are OT related intervention. However, the sentence ends with “…provide clients with a connection to the outside world.” Either change the word intervention or change the end of the sentence to the client have increased access to healthcare.

Response: We revised the end of the sentence as you suggested.

62 – Stating “alleviates social isolation” is a strong claim; the citation is a SR that reports “reducing social isolation”. Reference #9.

Response: We modified it to "reduces social isolation".

64 – 65 - “These technologies have been applied to various areas in occupational therapy, highlighting the effectiveness, perspective, and feasibility of ICT use.”  Need citation to prove this statement.

Response: The citation was added.

81 – 83 - In this regard, OTs play a key role in the therapeutic application of technology and recommendations for clients, and their roles evolve in parallel with technological advancements [18].  “their” refers to whom, there are two subjects (OTs or clients)? The end of the sentence is confusing.

Response: “their” refers to OTs, and it has been modified to “the roles of OTs” for clarity.

Results

 3.2. Error with the Likert scale

- Figure 1 A and B Likert scale are incorrect the # 5 and #1 both state “strongly agree”

Response: As suggested, we corrected it.

Discussion

 4.2. Familiarity with and use of ICT

263-264 – when referring to “this survey” drawing conclusion about “healthcare 2 professionals” is inaccurate, this survey was with OTs. Remove “by healthcare professionals.”

Response: As suggested, we revised it.

265 – 268 – “Despite the changes…” Awkward sentence

“…current domestic OT curriculum are not clearly integrated.” This survey did not address OT curriculum; therefore, this statement is not supported.

Response: We revised “current domestic OT curriculum” to “current domestic OT practice”. Although curriculum was not covered in the survey, unfamiliarity with ICT is associated with educational aspects such as lack of updated knowledge and clinical application. As you pointed out, rather than referring to the curriculum, I have modified it to be “continuing ICT education or ICT knowledge acquisition”. Please consider whether it does not interfere with the flow of discussion.

271 – 272 – “This will promote the dissemination of the latest knowledge and clinical cases of ICT use through continuing education.” Curriculum changes may lead to dissemination but necessarily through continuing education; this is not applicable link. Move everything that has to do with curriculum and continuing education to a closing paragraph to in this section.

Response: The following sentences (268-272) of 264 line have been rewritten. In relation to ICT familiarity, I think that the role of knowledge acquisition and continuous education is important. The mention of OT curriculum you mentioned has been deleted, and the need for education and knowledge acquisition opportunities for the latest knowledge and clinical application has been emphasized. Please consider this.

284 paragraph – This section is about 4.2. Familiarity with and use of ICT Begin with explicitly stating Client lack of experience with ICT is a barrier for use, for example the elderly. Confusing opening to this sentence (290) “Familiar users of this survey…”

Response: Before that sentence “Familiar users of this survey…”, additional sentences related to the acquisition of new information and leisure skills of the elderly were described.

294 (paragraph) – fold in familiarity language to be consistent with your sub-header. Recommend concluding 4.2. Familiarity with and use of ICT with a closing paragraph about curriculum and continuing education recommendations.

Response: We rewrote the sentences after line 268. Please confirm my response to ‘271 – 272’ what you commented. If you think it is not appropriate, I will reflect it in the 2nd revision.

327 – throughout the paper you have used client(s) do you want to use patients or clients here?

Response: Throughout the paper, patient(s) was changed to “client(s)”.

Conclusion: 362 – 363 - “In addition, it will be necessary to develop technology for commercialization and to reduce the price of accessible technology for clients in clinical practice.” This is outside of the scope of OT and making assumptions that commercialization will reduce price.

Response: The sentence you mentioned has been revised.

References Spacing layout of words, some lines are “justified”

Response: As suggested, we modified it.

* The revised parts were marked in red.

This manuscript is a resubmission of an earlier submission. The following is a list of the peer review reports and author responses from that submission.

Round 1

Reviewer 1 Report

The manuscript addresses an important topic that is key to transforming health care everywhere, particularly as the COVID 19 pandemic has increased use of internet health based capabilities.

It was  pleasure to review this manuscript that can serve as a model for replication in other countries.  I have minor recommendations for revision that include content clarification and minor editing that follow:

  1. In the first introductory paragraph, the authors refer to the Republic of Korea's aim to reduce medical costs and quality of medical care via the convergence of health care and ICT.  It may be appropriate to show how this is thematically related to aspects of the Triple Aim for Health Care in the US that shares two of these goals.
  2. In the first paragraph under the Method section, line 82, replace "with regard to" with as regards.
  3. Change the word utilize to use throughout the manuscript as may be needed.  
  4. Section 2.2 Participants - How was "more than one year of clinical experience considered skilled considered skilled in occupational therapy practice" measured or assessed.  Later in the paper some discussion related to this is "sort of" put forth, but I was not really clear.  I question if one year of practice constitutes skilled practice.  You may wish to revise this concept.
  5. Appropriate use of nonparametric statistical tests noted and appreciated.
  6. 4.4 Limitations section:  lines 321 and  322; I do not understand what the following phrase means:  "we referred to the ICT-related list proposed by domestic institutions and those that have been reviewed and pilot tested by several experts"  Please clarify as which institutions, how reviewed and how pilot tested is not clear to the reader.  It also should be considered in limitations to generalizability of the study.

Author Response

1. In the first introductory paragraph, the authors refer to the Republic of Korea's aim to reduce medical costs and quality of medical care via the convergence of health care and ICT.  It may be appropriate to show how this is thematically related to aspects of the Triple Aim for Health Care in the US that shares two of these goals.

Response: We revised it as suggested.

2. In the first paragraph under the Method section, line 82, replace "with regard to" with as regards.

Response: We revised it as suggested.

3. Change the word utilize to use throughout the manuscript as may be needed.  

Response: We revised it as suggested.

4. Section 2.2 Participants - How was "more than one year of clinical experience considered skilled considered skilled in occupational therapy practice" measured or assessed.  Later in the paper some discussion related to this is "sort of" put forth, but I was not really clear.  I question if one year of practice constitutes skilled practice.  You may wish to revise this concept.

Response: Based on the levels of professional expertise(novice, advanced beginner, competent, proficient, and expert) defined by Benner, novice therapists who are the first year of practice after graduation were excluded, and only therapists with a level of 'advanced beginner' (acceptable performance) or higher with more than 1 year of clinical experience were included.

We have described the above related sentences in the section 2.2 Participants .

5. Appropriate use of nonparametric statistical tests noted and appreciated.

Response: Thanks for your comment.

6. 4.4 Limitations section:  lines 321 and  322; I do not understand what the following phrase means:  "we referred to the ICT-related list proposed by domestic institutions and those that have been reviewed and pilot tested by several experts"  Please clarify as which institutions, how reviewed and how pilot tested is not clear to the reader.  It also should be considered in limitations to generalizability of the study.

Response: Thank you for your comments. In the Survey design section of the Method, it has already been described in detail from which institutions in Korea the ICT-related list was proposed, by whom the derived survey items were reviewed and revised, and which aspects of the survey items were pilot assessed. What you mentioned was more clearly described as Korea's domestic assistive technology institutions, and the derived survey items were reviewed, revised, and pilot tested by several experts, briefly summarized it in Limitations. Please refer to the Survey design section of the method, and if you think it is necessary to mention the details in Limitations, we will reflect them in the second revision.

 * Revised sentences, phrases, or words were highlighted.

Reviewer 2 Report

The manuscript entitled: "Use of Technology by South Korean Occupational Therapists Working in Hospitals: A Cross-Sectional Study " is poorly written and need a lot of revision.

The authors did not mention about the previous study in a correct correlated way.

No information was presented about other geographical regions.

In the introduction, ICT-based interventions and recommendations are not clear.

In the methods, the research question is not clear

In the methods, the data analysis is not sufficient.

In results, more variables are still needed to be included.

The discussion is weak and need to be rewritten.

Author Response

The manuscript entitled: "Use of Technology by South Korean Occupational Therapists Working in Hospitals: A Cross-Sectional Study " is poorly written and need a lot of revision.

Response: Thanks for your comments. We've tried to revise it as you suggested, but it may not be enough. We hope you will consider it generously. Could you be a little more specific about what you think needs additional revision?

The authors did not mention about the previous study in a correct correlated way.

Response: As suggested, we revised the description about previous studies in the introduction. : “Zonneveld et al. [12] demonstrated …”, “Collins [13] reported …”, “Other studies reported…”

No information was presented about other geographical regions.

Response: Six metropolitan cities in South Korea were included in each region of province of employment. Specifically, the survey item of the province of employment included neighboring metropolitan cities in each region (e.g. Gyeongsang (including Busan and Ulsan Metropolitan City)). It is common in surveys conducted in Korea to present the current administrative district as a response item including neighboring metropolitan cities.

In the introduction, ICT-based interventions and recommendations are not clear.

Response: As suggested, we additionally described ICT-based interventions and recommendations on page 2. : “The use of ICT …”, “ICT-based interventions, such as ...”, “OTs have used and recommended … ”

In the methods, the research question is not clear.

Response: We revised the survey questions in Methods to be clearer.

In the methods, the data analysis is not sufficient.

Response: Although we have tried to stick to the planned survey and data analysis, it may not be sufficient in your opinion. Please reconsider your decision regarding this study, and may I ask for your detailed opinion so that it can be a better article?

In results, more variables are still needed to be included.

Response: Thanks for your comments. We tried to describe the data analysis results and present them in tables and figures. May I ask what the more variables you mentioned specifically are? If you give us a comment in this regard, we will reflect it in the second revision.

The discussion is weak and need to be rewritten.

Response: We additionally described the sentences in the opinions on the sections of therapeutic uses of ICT and limitations of discussion. If it is determined that additional revisions are necessary, we will reflect them in the second revision.

* Revised sentences, phrases, or words were highlighted.

Reviewer 3 Report

Thank you for allowing me to review this article. This is an interesting manuscript with nice information to know. 
I would like to suggest a number of points for improvement:
- It is necessary to indicate the type of study being carried out as it is not referenced anywhere in the document. 
- In methodology, indicate the context (when the recruitment of participants was carried out, the period in which the survey was active, the need to make several calls, etc.). Indicate the sample size.
- In the discussion section, discuss the generalisability of the results. 
- The quality of Figure 1 is low and improvement is recommended.

Author Response

Thank you for allowing me to review this article. This is an interesting manuscript with nice information to know. 
I would like to suggest a number of points for improvement:

- It is necessary to indicate the type of study being carried out as it is not referenced anywhere in the document. 

Response: The study design was mentioned in the study design section of Methods. : “A cross-sectional online survey was conducted to identify … .”

- In methodology, indicate the context (when the recruitment of participants was carried out, the period in which the survey was active, the need to make several calls, etc.). Indicate the sample size.

Response: We further described your suggestions in the Participants section of the Method.

- In the discussion section, discuss the generalisability of the results. 

Response: Thank you for your comments. Although the nature of the extensive data collection of the survey study aims to generalize, this study is cautious about discussing the generalizability of the study results due to the small sample size. This has already been described in the limitations and suggested the need for future research. In the future, we will conduct additional research to discuss the generalizability of ICT use by domestic OTs. If you have additional or specific suggestions on this, we will reflect them in the second revision.

- The quality of Figure 1 is low and improvement is recommended.

Response: The resolution of Figure 1. was increased and inserted into the text.

* Revised sentences, phrases, or words were highlighted.

Round 2

Reviewer 2 Report

Thanks 

Author Response

 English language and style are fine/minor spell check required.

Response: Thanks for your comment. As suggested, we checked and corrected grammar and typos.
